# Weakly supervised learning for multi-organ adenocarcinoma classification in whole slide images

**Masayuki Tsuneki**○*, **Fahdi Kanavati**

Medmain Research, Medmain Inc., Akasaka, Chuo-ku, Fukuoka, Japan

* tsuneki@medmain.com

**Data Availability Statement:** The datasets generated during and/or analysed during the current study are not publicly available due to specific institutional requirements governing privacy protection but are available from the

## Abstract

The primary screening by automated computational pathology algorithms of the presence or absence of adenocarcinoma in biopsy specimens (e.g., endoscopic biopsy, transbronchial lung biopsy, and needle biopsy) of possible primary organs (e.g., stomach, colon, lung, and breast) and radical lymph node dissection specimen is very useful and should be a powerful tool to assist surgical pathologists in routine histopathological diagnostic workflow. In this paper, we trained multi-organ deep learning models to classify adenocarcinoma in biopsy and radical lymph node dissection specimens whole slide images (WSIs). We evaluated the models on five independent test sets (stomach, colon, lung, breast, lymph nodes) to demonstrate the feasibility in multi-organ and lymph nodes specimens from different medical institutions, achieving receiver operating characteristic areas under the curves (ROC-AUCs) in the range of 0.91 -0.98.

## Introduction

Adenocarcinoma is a type of carcinoma that has the propensity to differentiate into glandular, ductal, and acinar cells in several organs (e.g., stomach, colon, lung, and breast). According to the Global Cancer Statistics 2020 [1], number of new deaths and % of all sites for stomach, colon, lung, and breast cancers were as follows: 768,793 cases (7.7%) in stomach, 576,858 cases (5.8%) in colon, 1,796,144 cases (18.0%) in lung, and 684,996 cases (6.9%) in breast. Adenocarcinoma is the most common type of cancer affecting these four organs, so that adenocarcinoma classification in the primary organs especially on biopsy specimens is one of the most important histopathological inspection in clinical workflow to determine the strategies of cancer treatment. Moreover, lymph nodes are the most common site of metastatic adenocarcinoma, and can be constituted the first clinical manifestation of the cancer. The important clinical practice of the surgical pathologist is to identify the presence or absence of a malignant process in the lymph node. If cancer cells are identified within the efferent lymph vessels and extra-nodal tissues, it is necessary to note in the pathological report because of the possible prognostic significance. Histopathological evaluation of lymph node metastasis is very important for staging of tumors, documentation of tumor recurrence, and prediction of the most

corresponding author on reasonable request. The datasets that support the findings of this study are available from International University of Health and Welfare, Mita Hospital (Tokyo, Japan) and Kamachi Group Hospitals (Fukuoka, Japan), but restrictions apply to the availability of these data, which were used under a data use agreement which was made according to the Ethical Guidelines for Medical and Health Research Involving Human Subjects as set by the Japanese Ministry of Health, Labour and Welfare, and so are not publicly available. The data contains potentially sensitive information. However, the data are available from the authors upon reasonable request for private viewing and with permission from the corresponding medical institutions within the terms of the data use agreement and if compliant with the ethical and legal requirements as stipulated by the Japanese Ministry of Health, Labour and Welfare. Contact person: Professor Dr. Takayuki Shiomi, Department of Pathology, Faculty of Medicine, International University of Health and Welfare (Tokyo, Japan) phone: +81-476-20-7701 E-mail: t_shiomi@iuhw.ac.jp (2) Ethical board of Kamachi Group Hospitals (Wajiro, Shinkuki, Shinkomonji, andShinmizumaki Hospital) Contact person: Dr. Shigeo Nakano, Head of Department of Surgical Pathology at Kamachi Group Hospitals (Fukuoka, Japan) Phone: +81-92-608-0001 E-mail: sdnakano@harajuku-reha.com

**Funding:** This study is based on results obtained from a project, JPNP14012, subsidized by the New Energy and Industrial Technology Development Organization (NEDO). The founder provided support in the form of salaries for authors M.T. and F.K, but did not have any additional role in the study design, data collection and analysis, decision to publish, or preparation of the manuscript. The specific roles of these authors are articulated in the 'author contributions' section.

**Competing interests:** M.T. and F.K. are employees of Medmain Inc. (Fukuoka, Japan). This does not alter our adherence to PLOS ONE policies on sharing data and materials.

probable primary site for a metastatic cancer of uncertain primary site. However, in the routine practical diagnosis, frequently there are numerous number of lymph nodes to be inspected in a single glass slide and there are number of radical lymph node dissection specimen glass slides in the same patient, which should be a workload burden for surgical pathologists.

The incorporation of deep learning models in routine histopathological diagnostic workflow is on the horizon and is a promising technology, allowing the potential of reducing the burden of time-consuming diagnosis and increasing the detection rate of anomalies including cancers. Deep learning has been widely applied in tissue classification and adenocarcinoma detection on whole-slide images (WSIs), cellular detection and segmentation, and the stratification of patient outcomes [2–15]. Previous works have looked into applying deep learning models for adenocarcinoma classification separately for different organ, such as stomach [15–17], colon [15, 18], lung [16, 19], and breast [20, 21] histopathological specimen WSIs. Although these existing models exhibited very high ROC-AUCs for each organ, they cannot classify adenocarcinoma across organs accurately.

In this study, we trained deep learning models using weakly-supervised learning to predict adenocarcinoma in WSIs of stomach, colon, lung, and breast biopsy specimens for primary tumors as well as radical lymph node dissection specimens for metastatic carcinoma using training datasets for stomach, colon, lung, and breast biopsy specimen WSIs without annotations. We evaluated the models on each primary organ biopsy specimen (stomach, colon, lung, and breast) as well as radical lymph node dissection specimens to evaluate presence or absence of metastatic adenocarcinoma, achieving and ROC-AUC from 0.91 to 0.9 8. Our results suggest that deep learning algorithms might be useful for histopathological diagnostic aids for adenocarcinoma classification in primary organs and lymph node metastatic cancer screening.

## Materials and methods

### Clinical cases and pathological records

In the present retrospective study, a total of 8,896 H&E (hematoxylin & eosin) stained histopathological specimen slides of human adenocarcinoma and non-adenocarcinoma (adenoma and non-neoplastic) were collected from the surgical pathology files of five hospitals: International University of Health and Welfare (IUHW), Mita Hospital (Tokyo, Japan) and Kamachi Group Hospitals (total four hospitals: Wajiro, Shinkuki, Shinkomonji, and Shinmizumaki Hospital) (Fukuoka, Japan) after histopathological review by surgical pathologists. Adenoma cases were included as adenoma is a common differential diagnosis and exhibits some similarities to adenocarcinoma. The histopathological specimens were selected randomly to reflect a real clinical settings as much as possible. Prior to the experimental procedures, each WSI diagnosis was observed by at least two pathologists with the final checking and verification performed by senior pathologists. All WSIs were scanned at a magnification of x20 using the same Leica Aperio AT2 Digital Whole Slide Scanner (Leica Biosystems, Tokyo, Japan) and were saved as SVS file format with JPEG2000 compression.

### Dataset

Hospitals which provided histopathological specimen slides were anonymised by randomly assigning a letter (e.g., Hospital-A, B, C, D, and E). Table 1 breaks down the distribution of training sets from four domestic hospitals (Hospital-A, B, C, and D). Table 2 shows the distribution of 1K (1,000 WSIs), 2K (2,000 WSIs), and 4K (4,000 WSIs) training sets. Validation sets were selected randomly from the training sets and the numbers of validation sets were given in parentheses (Table 2). The distribution of test sets from five domestic hospitals (Hospital-A, B, C, D, and E) was summarized in Table 3. In both training and test sets, stomach, colon, lung,

**Table 1. Distribution of cases in the training sets obtained from different hospitals (A-D).**

| Organ | Specimen type | Class | Diagnosis | Hospital-A | Hospital-B | Hospital-C | Hospital-D | total |
|---|---|---|---|---|---|---|---|---|
| Stomach | Endoscopic biopsy | Adenocarcinoma | Adenocarcinoma | 250 | 100 | 150 | 0 | 500 |
| | | Non-adenocarcinoma | Adenoma | 40 | 20 | 10 | 0 | 70 |
| | | | Non-neoplastic | 200 | 100 | 130 | 0 | 430 |
| | | | total | 490 | 220 | 290 | 0 | 1000 |
| Colon | Endoscopic biopsy | Adenocarcinoma | Adenocarcinoma | 150 | 150 | 0 | 200 | 500 |
| | | Non-adenocarcinoma | Adenoma | 30 | 30 | 0 | 40 | 100 |
| | | | Non-neoplastic | 200 | 100 | 0 | 100 | 400 |
| | | | total | 380 | 280 | 0 | 340 | 1000 |
| Lung | TBLB | Adenocarcinoma | Adenocarcinoma | 200 | 100 | 0 | 100 | 400 |
| | | Non-adenocarcinoma | Non-neoplastic | 300 | 200 | 0 | 100 | 600 |
| | | | total | 500 | 300 | 0 | 200 | 1000 |
| Breast | Needle biopsy | Adenocarcinoma | Invasive ductal carcinoma | 200 | 100 | 100 | 0 | 400 |
| | | Non-adenocarcinoma | Non-neoplastic | 300 | 200 | 100 | 0 | 600 |
| | | | total | 500 | 300 | 200 | 0 | 1000 |
| | | | total | 1870 | 1100 | 490 | 540 | 4000 |

and breast WSIs solely consisted of biopsy (stomach and colon: endoscopic biopsy, lung: trans-bronchial lung biopsy (TBLB), breast: needle biopsy) specimens and lymph node WSIs consisted of radical dissection specimens (Tables 1–3). The distribution of lymph nodes using test sets were summatized in Table 4. All training sets WSIs were not manually annotated and the training algorithm only used the WSI labels which were extracted from the histopathological diagnostic reports after reviewing surgical pathologists; meaning that the only information available for the training was whether the WSI contained adenocarcinoma or non-adenocarcinoma but no information available about the location of the cancerous lesions.

## Deep learning models

In this study, we used the EfficientNetB1 [22] as the architecture of our models. We observed no further improvements from using larger models. We used the partial fine-tuning approach

**Table 2. Distribution of cases in the training sets and validation sets.** The numbers of validation cases are given in parentheses.

| Organ | Specimen type | Class | Diagnosis | 1K-training sets | 2K-training sets | 4K-training sets |
|---|---|---|---|---|---|---|
| Stomach | Endoscopic biopsy | Adenocarcinoma | Adenocarcinoma | 130 (8) | 250 (8) | 500 (8) |
| | | Non-adenocarcinoma | Adenoma | 20 (3) | 40 (3) | 70 (3) |
| | | | Non-neoplastic | 100 (4) | 210 (4) | 430 (4) |
| | | | total | 250 (15) | 500 (15) | 1000 (15) |
| Colon | Endoscopic biopsy | Adenocarcinoma | Adenocarcinoma | 130 (8) | 250 (8) | 500 (8) |
| | | Non-adenocarcinoma | Adenoma | 30 (3) | 50 (3) | 100 (3) |
| | | | Non-neoplastic | 90 (4) | 200 (4) | 400 (4) |
| | | | total | 250 (15) | 500 (15) | 1000 (15) |
| Lung | TBLB | Adenocarcinoma | Adenocarcinoma | 120 (8) | 200 (8) | 400 (8) |
| | | Non-adenocarcinoma | Non-neoplastic | 130 (7) | 300 (7) | 600 (7) |
| | | | total | 250 (15) | 500 (15) | 1000 (15) |
| Breast | Needle biopsy | Adenocarcinoma | Invasive ductal carcinoma | 120 (8) | 200 (8) | 400 (8) |
| | | Non-adenocarcinoma | Non-neoplastic | 130 (7) | 300 (7) | 600 (7) |
| | | | total | 250 (15) | 500 (15) | 1000 (15) |
| | | | total | 1000 (60) | 2000 (60) | 4000 (60) |

**Table 3. Distribution of cases in the test sets obtained from hospitals (A-E).**

| Organ | Specimen type | Class | Diagnosis | Hosp-A | Hosp-B | Hosp-C | Hosp-D | Hosp-E | total |
|---|---|---|---|---|---|---|---|---|---|
| Stomach | Endoscopic biopsy | Adenocarcinoma | Adenocarcinoma | 108 | 120 | 57 | 0 | 52 | 337 |
| | | Non-adenocarcinoma | Adenoma | 33 | 35 | 27 | 0 | 32 | 127 |
| | | | Non-neoplastic | 263 | 86 | 78 | 0 | 109 | 536 |
| | | | total | 404 | 241 | 162 | 0 | 193 | 1000 |
| Colon | Endoscopic biopsy | Adenocarcinoma | Adenocarcinoma | 125 | 158 | 0 | 74 | 42 | 399 |
| | | Non-adenocarcinoma | Adenoma | 61 | 55 | 0 | 83 | 78 | 277 |
| | | | Non-neoplastic | 136 | 109 | 0 | 62 | 17 | 324 |
| | | | total | 322 | 322 | 0 | 219 | 137 | 1000 |
| Lung | TBLB | Adenocarcinoma | Adenocarcinoma | 211 | 156 | 0 | 103 | 0 | 470 |
| | | Non-adenocarcinoma | Non-neoplastic | 259 | 198 | 0 | 73 | 0 | 530 |
| | | | total | 470 | 354 | 0 | 176 | 0 | 1000 |
| Breast | Needle biopsy | Adenocarcinoma | Invasive ductal carcinoma | 289 | 44 | 59 | 0 | 0 | 392 |
| | | Non-adenocarcinoma | Non-neoplastic | 233 | 166 | 177 | 0 | 0 | 576 |
| | | | total | 522 | 210 | 236 | 0 | 0 | 968 |
| Lymph node | Radical dissection | Adenocarcinoma | Adenocarcinoma | 57 | 79 | 10 | 0 | 0 | 146 |
| | | Non-adenocarcinoma | Non-neoplastic | 222 | 314 | 246 | 0 | 0 | 782 |
| | | | total | 279 | 393 | 256 | 0 | 0 | 928 |
| | | | total | 1997 | 1520 | 654 | 395 | 330 | **4896** |

[23] to train them. This method starts with an existing pre-trained models on ImageNet and fine-tunes only the affine parameters of the batch normalization layers and the final classification layer while leaving the remaining weights frozen. Fig 1 shows an overview of the training method.

As we only had WSI labels, we used a weakly supervised method to train the models. The training method is similar to the one described in [24].

WSIs typically have large areas of white background that is not required for training the model and can easily be eliminated with preprocessing via thresholding using Otsu's method [25]. This creates a mask of the tissue regions from which it would then be possible to sample tiles in real-time using the OpenSlide library [26] by providing coordinates from the tissue regions.

For a given WSI, we obtained a single prediction on the slide-level using the following approach: we divided the WSIs into a grid with a fixed stride, and we applied the model in a sliding window fashion over the grid, resulting in a predictions for the entire tissue regions.

**Table 4. Distribution of whole slide images (WSIs) in the lymph nodes test sets.**

| Resected organ | Clinical diagnosis | Histopathological diagnosis | WSI |
|---|---|---|---|
| Stomach | Advanced gastric cancer | Adenocarcinoma | 18 |
| | | Non-neoplastic | 97 |
| Colon | Advanced colon cancer | Adenocarcinoma | 21 |
| | | Non-neoplastic | 166 |
| Lung | Lung cancer | Adenocarcinoma | 38 |
| | | Non-neoplastic | 181 |
| Lung | Metastatic colon cancer | Adenocarcinoma | 27 |
| | | Non-neoplastic | 172 |
| Breast | Breast cancer | Invasive ductal carcinoma | 42 |
| | | Non-neoplastic | 166 |

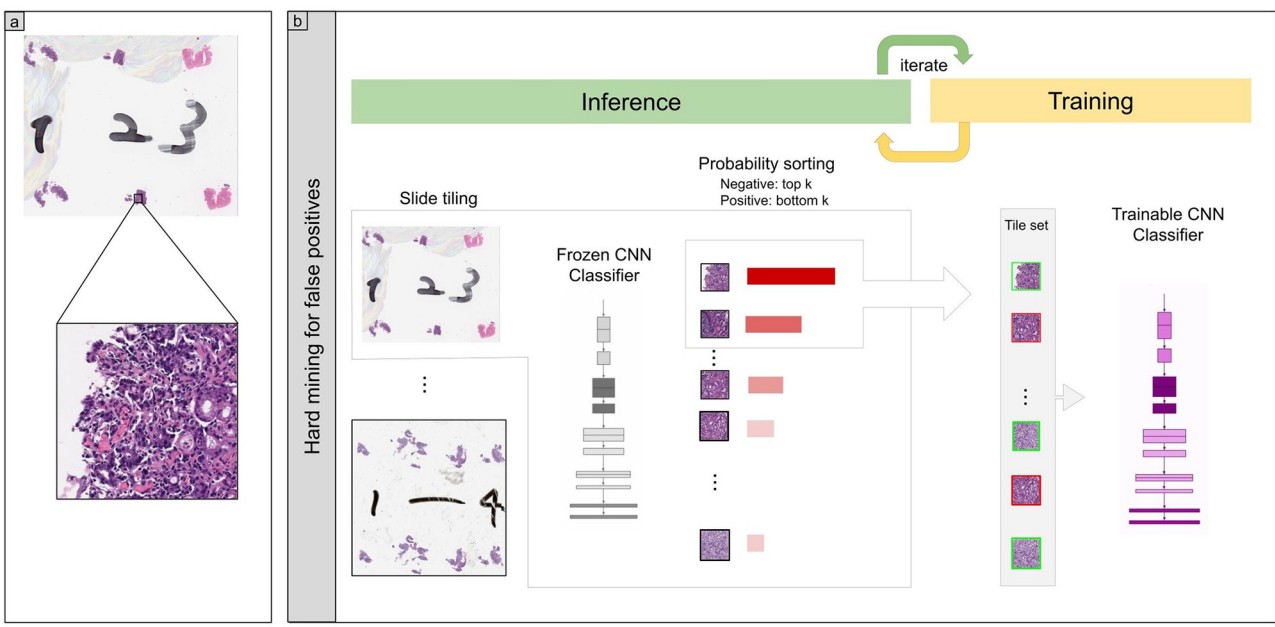

**Fig 1. Overview of training method.** (a) shows a zoomed-in example of a tile from a WSI. (b) During training, we alternated between an inference step and a training step. During the inference step, the model weights were frozen and the model was used to select tiles with the highest probability after applying it on the entire tissue regions of each WSI. The top k tiles with the highest probabilities were then selected from each WSI and placed into a queue. During training, the selected tiles from multiple WSIs formed a training batch and were used to train the model.

We then took the maximum probability from all the tiles as used that as a slide-level probability of the WSI having ADC.

During training, we initially performed a balanced random sampling of tiles from the tissue regions for first two epochs; this meant that we alternated between a positive WSI and a negative WSI and selecting an equal number of tiles from each. After the second epoch, we switched into hard mining of tiles, whereby we alternated between a positive WSI and a negative WSI; however, this time performing a sliding window inference on the entire tissue regions and then selecting the top $k$ tiles with the highest probabilities for being positive. If the WSI is negative, this effectively selects the tiles most likely to be false positives. The selected tiles were placed in a training subset, and once that subset contained $N$ tiles, a training was run whereby the model weights get updated. We used $k = 8$, $N = 256$, and a batch size of 32.

In addition, during training, we performed data augmentation of the images by performing random shifts in brightness, contrast, hue and saturation, and rotation angles as well as horizontal and vertical flipping.

We optimised the model weights by minimising the binary cross-entropy loss using the Adam optimization algorithm [27] with the following parameters: $beta_1 = 0.9$, $beta_2 = 0.999$ and a learning rate of 0.001. We applied a learning rate decay of 0.95 every 2 epochs. We used early stopping by tracking the performance of the model on a validation set; this allows stopping the training when no improvement was observed for more than 10 epochs. The model with the lowest validation loss was chosen as the final model.

## Software and statistical analysis

The deep learning models were implemented and trained using TensorFlow [28]. AUCs were calculated in python using the scikit-learn package [29] and plotted using matplotlib [30]. The 95% CIs of the AUCs were estimated using the bootstrap method [31] with 1000 iterations.

The true positive rate (TPR) was computed as

$$TPR = \frac{TP}{TP + FN} \tag{1}$$

and the false positive rate (FPR) was computed as

$$FPR = \frac{FP}{FP + TN} \tag{2}$$

Where TP, FP, and TN represent true positive, false positive, and true negative, respectively. The ROC curve was computed by varying the probability threshold from 0.0 to 1.0 and computing both the TPR and FPR at the given threshold.

## Compliance with ethical standards

The experimental protocol was approved by the ethical board of International University of Health and Welfare (No. 19-Im-007) and Kamachi Group Hospitals (No. 173). All research activities complied with all relevant ethical regulations and were performed in accordance with relevant guidelines and regulations in the all hospitals mentioned above.

## Availability of data and material

The datasets generated during and/or analysed during the current study are not publicly available due to specific institutional requirements governing privacy protection but are available from the corresponding author on reasonable request. The datasets that support the findings of this study are available from International University of Health and Welfare, Mita Hospital (Tokyo, Japan) and Kamachi Group Hospitals (Fukuoka, Japan), but restrictions apply to the availability of these data, which were used under a data use agreement which was made according to the Ethical Guidelines for Medical and Health Research Involving Human Subjects as set by the Japanese Ministry of Health, Labour and Welfare, and so are not publicly available. The data contains potentially sensitive information. However, the data are available from the authors upon reasonable request for private viewing and with permission from the corresponding medical institutions within the terms of the data use agreement and if compliant with the ethical and legal requirements as stipulated by the Japanese Ministry of Health, Labour and Welfare.

## Results

### Insufficient AUC performance of WSI adenocarcinoma evaluation using existing stomach adenocarcinoma classification model

Prior to the training of multi-organ adenocarcinoma model, we have demonstrated the existing stomach adenocarcinoma classification model [15] AUC performance on test sets (Table 3). Table 5 and Fig 2A show that stomach and colon endoscopic biopsy WSIs exhibited high ROC-AUC and low log loss values but not in lung TBLB, breast needle biopsy, and radical lymph node dissection WSIs. Thus, we have trained the models using different WSI number of training sets (Table 2).

### High AUC performance of WSI evaluation of adenocarcinoma histopathology images

We trained models using weakly-supervised (WS) learning which could be used with weak labels (WSI labels) [24]. We trained using the EfficientNetB1 convolutional neural network

**Table 5. ROC-AUC and log loss results for adenocarcinoma classification on test sets using existing stomach adenocarcinoma classification model.**

| test sets | Existing stomach adenocarcinoma model | |
| --- | --- | --- |
| | ROC-AUC | log loss |
| Stomach endoscopic biopsy | 0.937 [0.918–0.953] | 0.450 [0.364–0.557] |
| Colon endoscopic biopsy | 0.986 [0.977–0.992] | 0.192 [0.142–0.252] |
| Lung TBLB | 0.698 [0.665–0.726] | 1.807 [1.680–1.960] |
| Breast needle biopsy | 0.888 [0.864–0.907] | 1.225 [1.111–1.329] |
| Lymph node radical dissection | 0.804 [0.771–0.832] | 1.940 [1.787–2.091] |

(CNN) architecture at magnification x10. The models were applied in a sliding window fashion with input tiles of 224x224 pixels and a stride of 256 (Fig 1). To train the deep learning models, we used a total of 1,000 (1K), 2,000 (2K), and 4,000 (4K) training set WSIs (Table 2). This resulted in three different models: (1) WS-1K: 224, x10 EfficientNetB1, (2) WS-2K: 224, x10 EfficientNetB1, and (3) WS-4K: 224, x10 EfficientNetB1. We evaluated the models on test sets from domestic hospitals (Table 3). For each test set (stomach endoscopic biopsy, colon endoscopic biopsy, lung TBLB, breast needle biopsy, and radical lymph node dissection), we computed the ROC-AUC, log loss, accuracy, sensitivity, and specificity (using a probability threshold of 0.5) and summarized the results in Tables 6 and 7 and Fig 2B–2D. The models trained using 2K and 4K training sets have a higher ROC-AUCs compared to the model trained using 1K and existing stomach adenocarcinoma model (Table 6, Fig 2). However, there was no obvious difference between the model trained using 2K and 4K training sets (Table 6, Fig 2C and 2D). In test sets from domestic hospitals, the model (WS-4K: 224, x10 EfficientNetB1) achieved very high ROC-AUCs (0.912–0.97 8) with low values of log loss (0.203–0.437) (Table 6). In all test sets, the model (WS-4K: 224, x10 EfficientNetB1) achieved very high accuracy (0.853–0.9 29), sensitivity (0.79 6–0.9 11), and specificity (0.82 5– 0.931) (Table 7). As shown in Fig 2, Tables 5–7, the model (WS-4K: 224, x10 EfficientNetB1) is fully applicable for multi-organ adenocarcinoma classification in wide variety of organs (stomach, colon, lung, breast, and lymph node) WSIs. Figs 3–7 show representative cases of true positive, true-negative, false positive, and false negative, respectively from using the model (WS-4K: 224, x10 EfficientNetB1).

## True positive adenocarcinoma prediction of stomach, colon, lung, and breast biopsy WSIs

Our model (WS-4K: 224, x10 EfficientNetB1) satisfactorily predicted adenocarcinoma in stomach endoscopic biopsy (Fig 3A–3C), colon endoscopic biopsy (Fig 3D–3G), lung TBLB (Fig 3H and 3I), and breast needle biopsy (Fig 3J and 3K) specimens. Importantly, the heatmap images showed true negative predictions of internal non-neoplastic tissue fragments (#2 in Fig 3A and 3B; #3 in Fig 3D and 3E; 3H–3K) which were confirmed by surgical pathologists.

## True positive adenocarcinoma prediction of radical lymph node dissection (lymphadenectomy) WSIs

A lymphadenectomy (radical lymph node dissection) is a surgical procedure to evaluate evidence of metastatic cancer. In routine histopathological diagnosis, the histopathological inspection of lymph nodes is one of the very important but time-consuming task to avoid the risk of medical oversight. Therefore, in clinical settings, the multi-organ adenocarcinoma model is more useful when performing histopathological diagnosis of lymphadenectomy

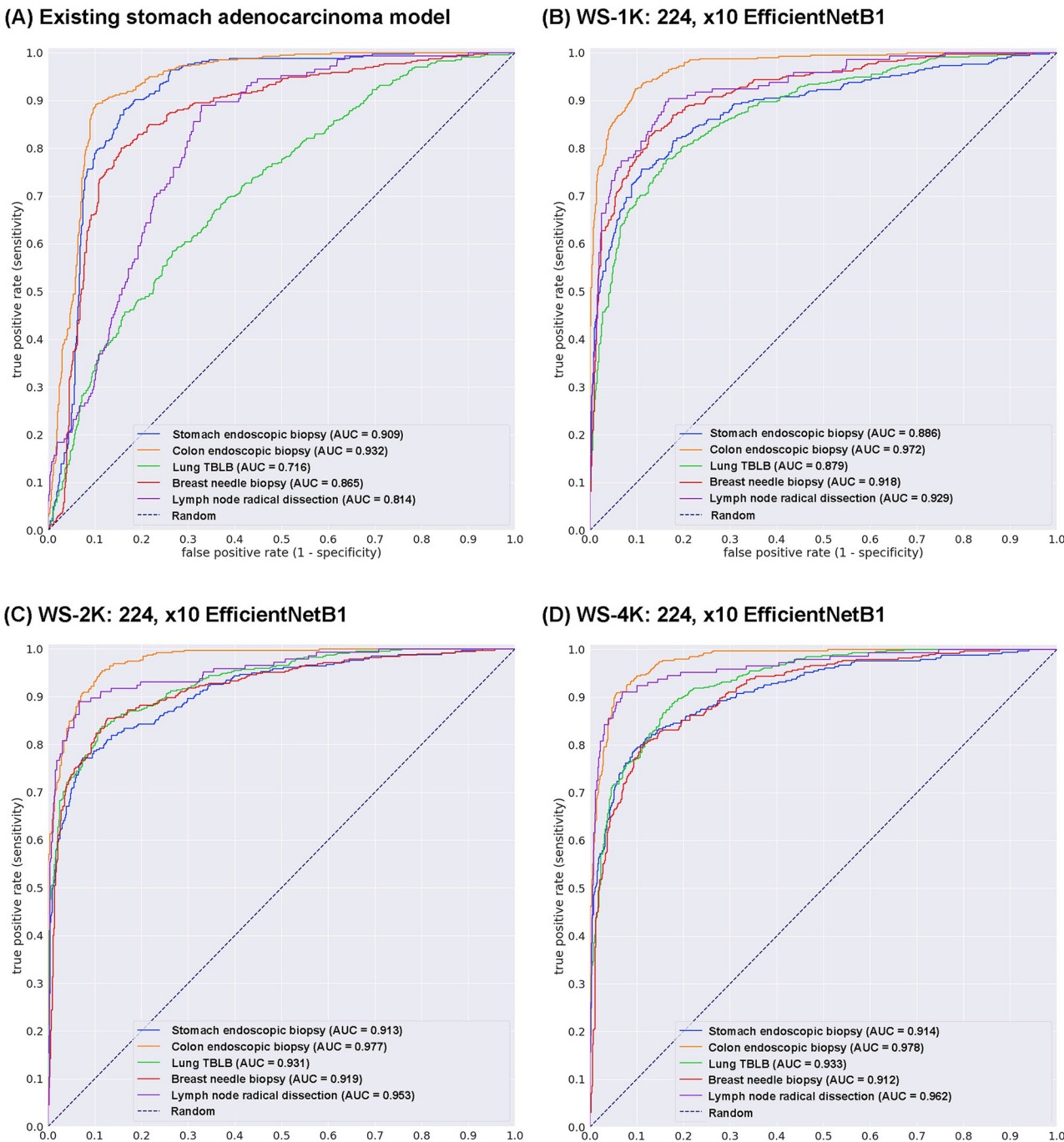

**Fig 2. ROC curves with AUCs from four different models (A-D) on the test sets: (A) Existing stomach adenocarcinoma classification model and weakly supervised (WS) learning models based on 1K (B), 2K (C), and 4K (D) training sets with tile size 224 px and magnification at x10.**

specimen WSIs. Our model (WS-4K: 224, x10 EfficientNetB1) perfectly predicted metastatic lung adenocarcinoma (Fig 4A–4D) and breast invasive ductal carcinoma (Fig 4E–4J). The heatmap images showed true negative predictions (Fig 4B) of internal non-neoplastic lymph nodes (Fig 4A). Importantly, adenocarcinoma localization areas in both metastatic lung

**Table 6. ROC-AUC and log loss results for adenocarcinoma classification on test sets using trained models.**

| | WS-1K: 224, x10 EfficientNetB1 | |
|---|---|---|
| test sets | ROC-AUC | log loss |
| Stomach endoscopic biopsy | 0.886 [0.864–0.912] | 0.415 [0.356–0.473] |
| Colon endoscopic biopsy | 0.973 [0.964–0.981] | 0.209 [0.175–0.242] |
| Lung TBLB | 0.879 [0.859–0.900] | 0.501 [0.443–0.555] |
| Breast needle biopsy | 0.919 [0.900–0.935] | 0.358 [0.315–0.404] |
| Lymph node radical dissection | 0.929 [0.903–0.951] | 0.427 [0.380–0.486] |
| | WS-2K: 224, x10 EfficientNetB1 | |
| test sets | ROC-AUC | log loss |
| Stomach endoscopic biopsy | 0.913 [0.894–0.932] | 0.351 [0.301–0.396] |
| Colon endoscopic biopsy | 0.977 [0.969–0.936] | 0.197 [0.167–0.226] |
| Lung TBLB | 0.931 [0.915–0.946] | 0.342 [0.300–0.386] |
| Breast needle biopsy | 0.919 [0.901–0.936] | 0.371 [0.325–0.423] |
| Lymph node radical dissection | 0.953 [0.939–0.978] | 0.228 [0.188–0.257] |
| | WS-4K: 224, x10 EfficientNetB1 | |
| test sets | ROC-AUC | log loss |
| Stomach endoscopic biopsy | 0.914 [0.890–0.931] | 0.355 [0.315–0.404] |
| Colon endoscopic biopsy | 0.978 [0.970–0.984] | 0.203 [0.173–0.236] |
| Lung TBLB | 0.933 [0.917–0.946] | 0.437 [0.391–0.494] |
| Breast needle biopsy | 0.912 [0.894–0.930] | 0.374 [0.330–0.421] |
| Lymph node radical dissection | 0.962 [0.942–0.978] | 0.309 [0.272–0.356] |

adenocarcinoma (Fig 4C) and breast invasive ductal carcinoma (Fig 4G) are positively predicted by heatmap images (Fig 4D and 4H).

## True negative adenocarcinoma prediction of radical lymph node dissection (lymphadenectomy) WSIs

Our model (WS-4K: 224, x10 EfficientNetB1) showed true negative predictions of metastatic adenocarcinoma in lymph nodes without evidence of cancer metastasis (Fig 5). In Fig 5A, there were numbers of lymph nodes with broad ranging of size (small to large) and shape (round to irregular) which were not predicted as metastatic lymph nodes (Fig 5B). Moreover, in Fig 5C, the lymph node was enlarged due to lymphadenitis (Fig 5E) but without evidence of metastatic adenocarcinoma which were not predicted as metastatic lymph nodes (Fig 5D).

## False positive adenocarcinoma prediction of radical lymph node dissection (lymphadenectomy) WSIs

Histopathologically, Fig 6A shows no evidence of metastatic adenocarcinoma. Our model (WS-4K: 224, x10 EfficientNetB1) exhibited false positive predictions of adenocarcinoma (Fig

**Table 7. Scores of accuracy, sensitivity, and specificity on test sets using the best model (WS-4K: 224, x10 EfficientNetB1).**

| | WS-4K: 224, x10 EfficientNetB1 | | |
|---|---|---|---|
| test sets | Accuracy | Sensitivity | Specificity |
| Stomach endoscopic biopsy | 0.859 [0.837–0.877] | 0.813 [0.766–0.850] | 0.882 [0.856–0.905] |
| Colon endoscopic biopsy | 0.929 [0.912–0.944] | 0.907 [0.878–0.935] | 0.943 [0.924–0.960] |
| Lung TBLB | 0.853 [0.831–0.875] | 0.885 [0.861–0.915] | 0.825 [0.792–0.855] |
| Breast needle biopsy | 0.853 [0.831–0.876] | 0.796 [0.755–0.837] | 0.892 [0.868–0.917] |
| Lymph node radical dissection | 0.928 [0.912–0.944] | 0.911 [0.866–0.955] | 0.931 [0.913–0.948] |

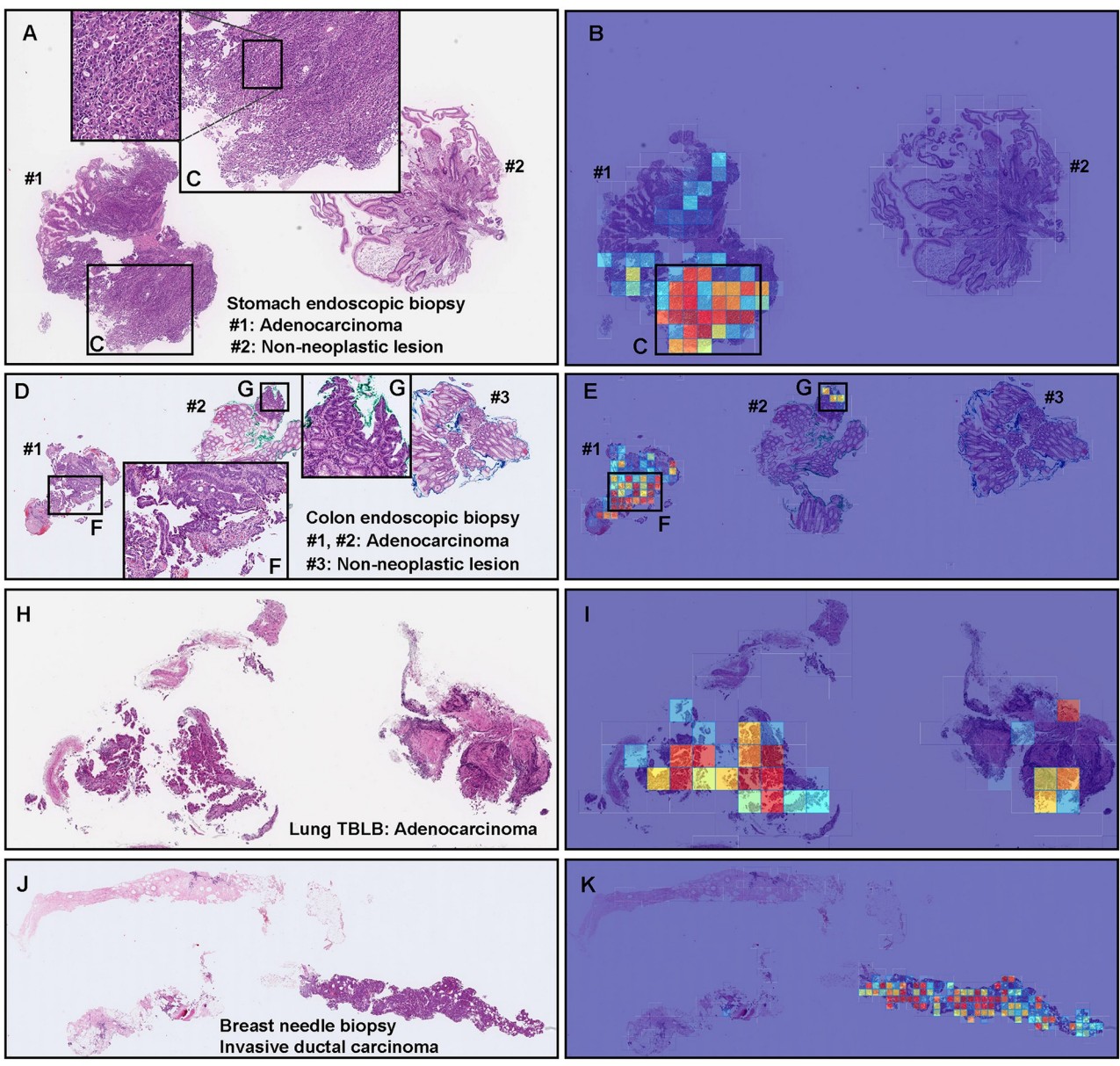

**Fig 3. Representative true positive adenocarcinoma classification of stomach, colon, lung, and breast biopsy test cases using the model (WS-4K: 224, x10 EfficientNetB1).** In the adenocarcinoma whole slide images (WSIs) of stomach endoscopic biopsy (A), colon endoscopic biopsy (D), lung transbronchial lung biopsy (TBLB) (H), and breast core needle biopsy (J) specimens, the heatmap images show true positive prediction of adenocarcinoma cells (B, E, F, G, I, K) which correspond respectively to H&E histopathology (A, C, D, F, G, H, J). The heatmap images show true negative predictions of non-neoplastic tissue fragments (#2 in (B) and #3 in (E)) and true positive predictions of adenocarcinoma tissue fragments (#1 in (B) and #1-#2 in (E)) which correspond respectively to H&E histopathology of adenocarcinoma area (C, F, G). The heatmap uses the jet color map where blue indicates low probability and red indicates high probability.

6B, 6D and 6F). These tissue areas (Fig 6C and 6E) showed dense hematoxylic artifacts induced by crushing during specimen handling procedures which could be the primary cause of false positive due to its morphological similarity to irregular shaped and dense nuclei in adenocarcinoma cells.

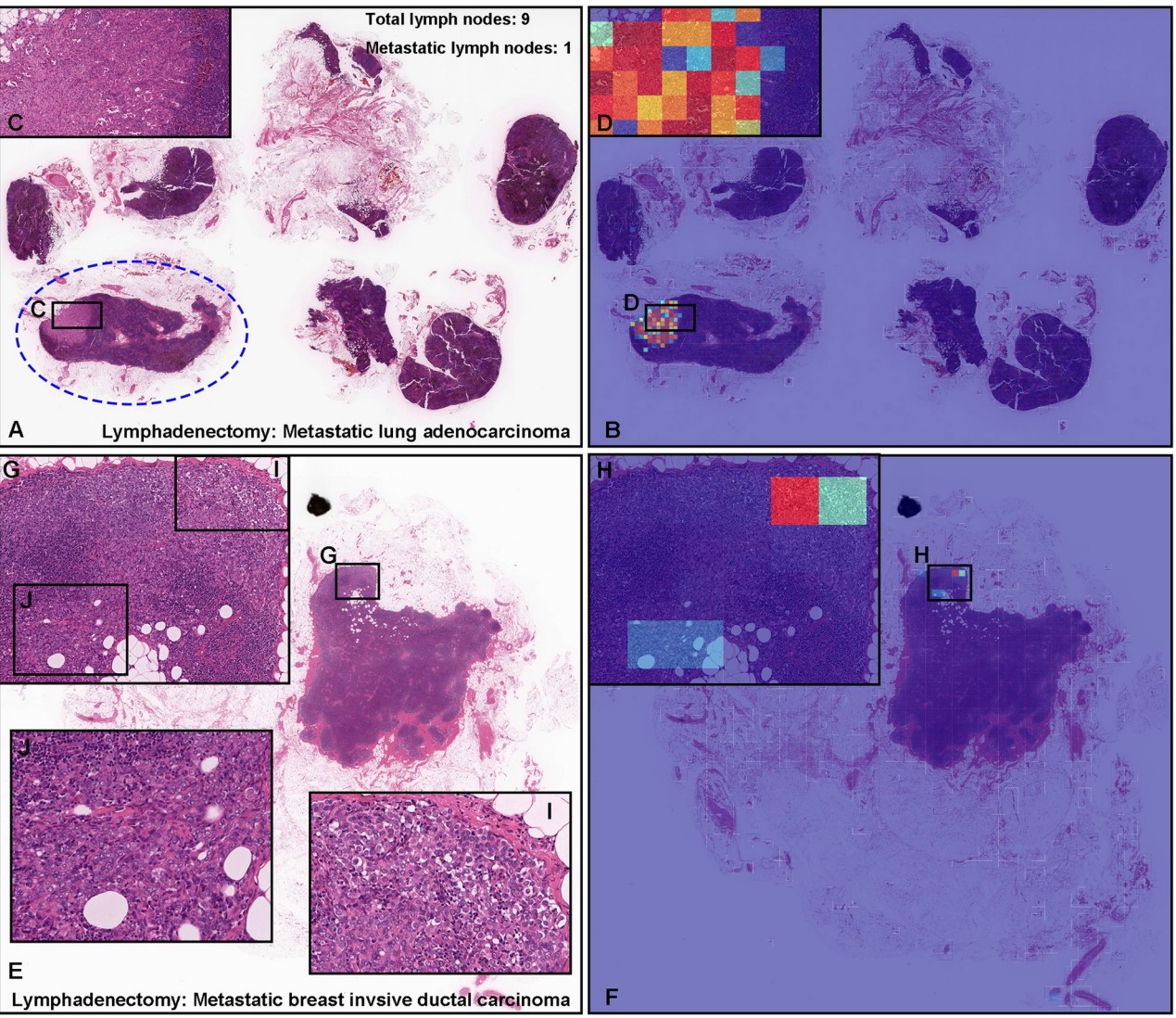

**Fig 4. Representative examples of metastatic adenocarcinoma true positive prediction outputs on cases from radical lymph node dissection (lymphadenectomy) test sets using the model (WS-4K: 224, x10 EfficientNetB1).** In the metastatic lung adenocarcinoma (A) and breast invasive ductal carcinoma (E) whole slide images (WSIs) of radical lymph node dissection specimens, the heatmap images show true positive prediction of metastatic lung adenocarcinoma (B, D) and breast invasive ductal carcinoma (F, H) cells which correspond respectively to H&E histopathology (A, C, E, G, I, J). According to the histopathological diagnostic report, in (A), only one lymph node (blue dot line circled) was positive for metastatic lung adenocarcinoma (C). The heatmap image (B) shows true positive prediction which was consistent with areas of metastatic lung adenocarcinoma invasion in the same lymph node (D). The heatmap image (B) also shows no positive predictions in the lymph nodes without evidence of cancer metastasis (A). As compared to (A), histopathologically, it was not easy to determine metastatic cancer areas in (E) at low power view. According to the histopathological report, in (E), metastatic breast invasive ductal carcinoma was localized in (G). The heatmap image (F) shows true positive predictions in (H) which are coincided with metastatic carcinoma infiltrating areas (G, I, J). The heatmap uses the jet color map where blue indicates low probability and red indicates high probability.

## False negative adenocarcinoma prediction of radical lymph node dissection (lymphadenectomy) WSIs

In Fig 7A, histopathologically, only two metastatic colon adenocarcinoma foci were observed in the left-most lymph node (Fig 7C). After double checking two independent pathologists, there were no more metastatic adenocarcinoma cells in Fig 7A. However, the heatmap image did not predict any adenocarcinoma cells (Fig 7B).

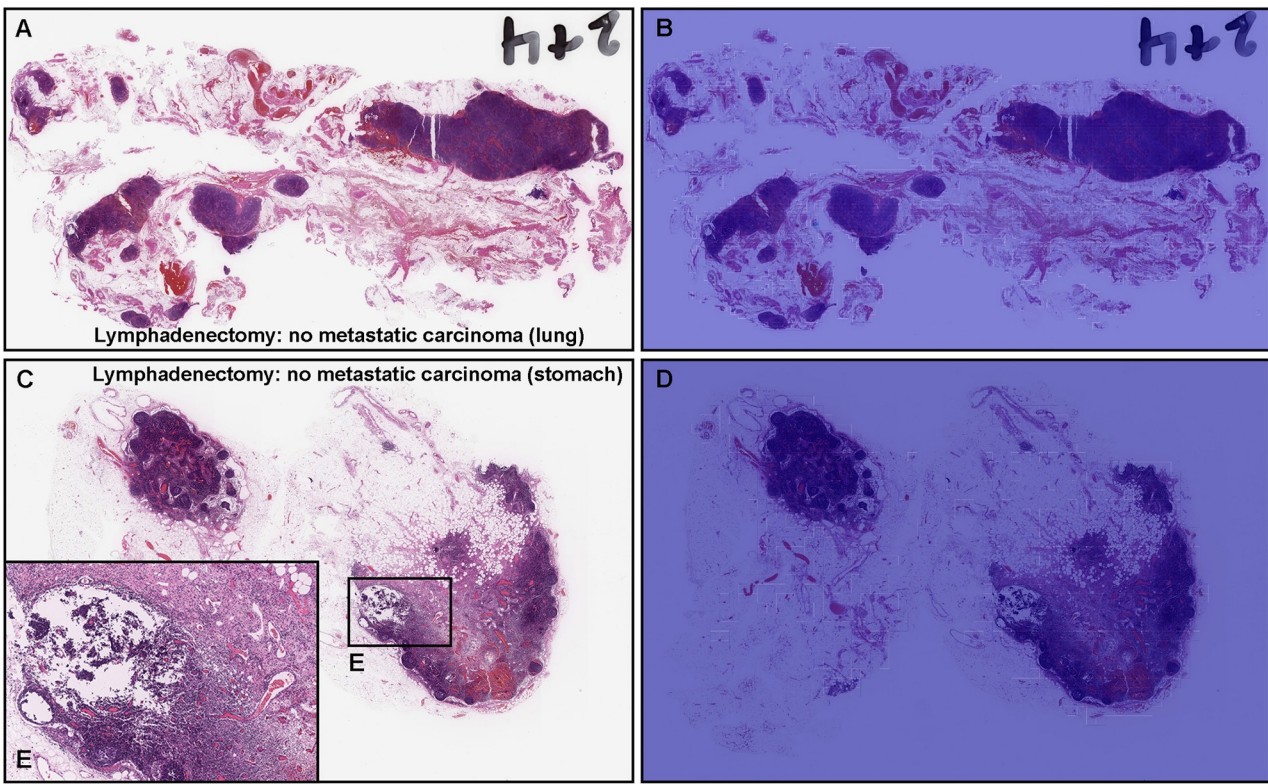

**Fig 5. Representative true negative metastatic adenocarcinoma classification of radical lymph node dissection (lymphadenectomy) test sets using the model (WS-4K: 224, x10 EfficientNetB1).** Histopathologically, in (A), there were diverse size (small to large) and shape (round to irregular) of lymph nodes without evidence of metastatic adenocarcinoma. The heatmap image (B) shows true negative prediction of metastatic adenocarcinoma. Histopathologically, in (C), there were lymph nodes with lymphadenitis (E) but without evidence of metastatic adenocarcinoma (C, E). The heatmap image (D) shows true negative prediction of metastatic adenocarcinoma. The heatmap uses the jet color map where blue indicates low probability and red indicates high probability.

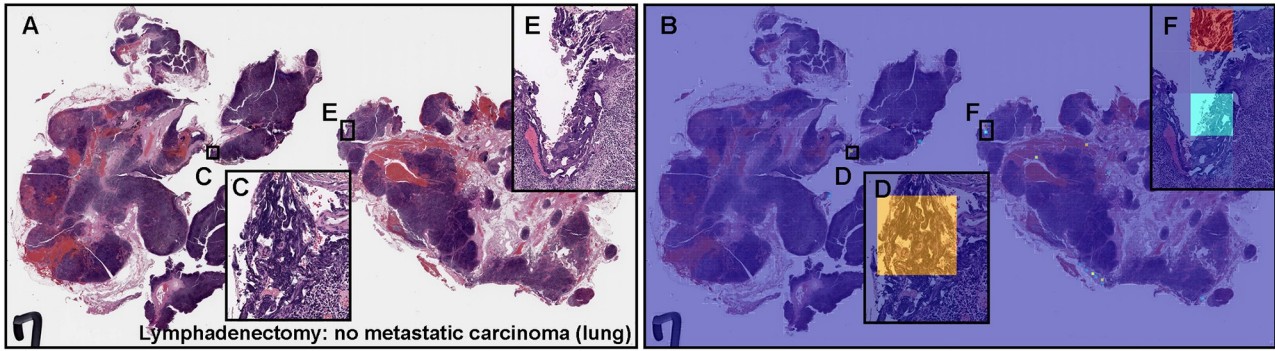

**Fig 6. Representative example of metastatic adenocarcinoma false positive prediction outputs on a case from the radical lymph node dissection (lymphadenectomy) test set using the model (WS-4K: 224, x10 EfficientNetB1).** Histopathologically, (A) has no sign of metastatic adenocarcinoma. The heatmap image (B) exhibits false positive predictions of adenocarcinoma (D, F) where the tissue consists of dense hematoxylic artifacts induced by crushing during specimen handling procedures (C, E), which most likely is the primary cause of the false positive prediction due to its morphological similarity to adenocarcinoma cells with irregular shaped and dense nuclei. The heatmap uses the jet color map where blue indicates low probability and red indicates high probability.

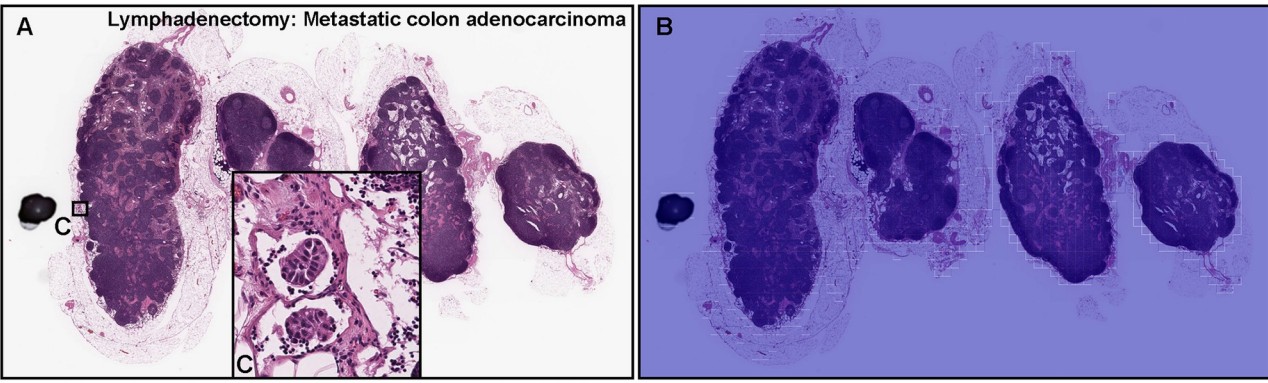

**Fig 7. Representative example of metastatic adenocarcinoma false negative prediction output on a case from the radical lymph node dissection (lymphadenectomy) test set using the model (WS-4K: 224, x10 EfficientNetB1).** According to the histopathological diagnostic report, this case (A) has metastatic adenocarcinoma foci in (C) but not in other areas. The heatmap image exhibited no positive adenocarcinoma prediction (B). The heatmap uses the jet color map where blue indicates low probability and red indicates high probability.

## Discussion

In the present study, we trained multi-organ deep learning models for the classification of adenocarcinoma in WSIs using weakly-supervised learning. The models were trained on WSIs obtained from four medical institutions and were then applied on multi-organ test sets obtained from five medical institutions to demonstrate the generalisation of the model on unseen data. The deep learning model (WS-4K: 224, x10 EfficientNetB1) achieved ROC-AUCs in the range of 0.91-0.9 8.

So far, we have been investigating adenocarcinoma classification on histopathological WSIs in diverse organs (e.g., stomach [15–17], colon [15, 18], lung [19, 24], and breast [20, 21]). These models are specific to each organ, and versatile adenocarcinoma histopathological classification model(s) which can be applied in multi-organ have not been developed to date. The global adenocarcinoma classification model in multi-organ may play key roles in first-screening processes especially radical lymph node dissection specimens which consist of a large number of lymph nodes in a single WSI in routine pathological diagnosis in the clinical laboratories.

Prior to the training, we have demonstrated the versatility of the existing models. For example, the existing stomach adenocarcinoma classification model [15] exhibited scores of high ROC-AUC and low log loss for the stomach and colon endoscopic biopsy test sets, but not for the lung, breast, and lymph node test sets (Table 5). Therefore, we have trained the deep learning models from scratch by the weakly-supervised learning approach in this study.

We have collected histopathological H&E stained specimens from as many medical institutions as possible to ensure diversities of histopathological variability and specimen quality in training sets (Table 1). In the training sets, we did not include radical lymph node dissection specimens because we would like to train the model based on the primary organs and predict metastatic adenocarcinoma in lymph nodes. In all training sets (1K, 2K, and 4K), WSIs from each organ (stomach, colon, lung, and breast) were equally distributed (Table 2).

In this study, we showed that it was possible to exploit the use of a moderate size training sets of 2,000 (2K) and 4,000 (4K) WSIs to train deep learning models using a weakly-supervised learning, and we have obtained high ROC-AUC performance on primary organ (stomach, colon, lung, and breast) and radical lymph node dissection test sets, which is highly promising in terms of the generalisation performance of our models to classify adenocarcinoma in multi-organs. Using the weakly-supervised learning method allowed us to train on

our datasets and obtain high performance without manually performed annotations. This means that it is possible to train a very high performance model for any type of cancer classification in multi-organ without having to have detailed cellular level or rough annotations or requiring an extremely large number of WSI. We have demonstrated the usefulness of weakly-supervised learning approach for lung carcinoma classification [24]. Importantly, there were no significant difference in ROC-AUC and log loss results between 2K and 4K training sets, meaning that small number (total 2,000 WSIs) of training datasets were enough for adenocarcinoma classification in multi-organ.

Our model satisfactorily predicted adenocarcinoma areas not only in primary organs (stomach, colon, lung, and breast) (Fig 3) but also in radical lymph node dissection specimens (Fig 4). In routine histopathological diagnosis, inspecting cancer metastasis in lymph nodes is laborious because usually there are a lot of lymph nodes with wide variety of sizes and shapes in glass slides. Our model can localise the prediction of adenocarcinoma invasion and visualise them as heatmap images (Fig 4) which would be a great tool for primary screening or double-check purpose in clinical workflow in laboratories. Importantly, our model can evaluate adenocarcinoma-free (non-metastatic) lymph nodes (Fig 5) which reflected high specificity (0.931) (Table 7). This is an important finding to apply our model in clinical workflow. This study is not without limitations. One limitation is the use of a single scanner type for the majority of collected cases. Another limitation is the presence of false positive/negatives. The false positives seem to be primarily caused by the dense hematoxylic artifacts induced by crushing during specimen handling procedures which have morphological similarities to adenocarcinoma cell clusters with irregular shaped and dense nuclei (Fig 6). Another major limitation is that the models were not validated in independent cohorts from different institutions.

## Acknowledgments

We are grateful for the support provided by Dr. Shin Ichihara at Department of Surgical Pathology, Sapporo Kosei General Hospital (Sapporo, Japan); Dr. Makoto Abe at Department of Pathology, Tochigi Cancer Center (Tochigi, Japan); Dr. Shigeo Nakano at Kamachi Group Hospitals (Fukuoka, Japan); Professor Takayuki Shiomi at Department of Pathology, Faculty of Medicine, International University of Health and Welfare (Tokyo, Japan); Dr. Ryosuke Matsuoka at Diagnostic Pathology Center, International University of Health and Welfare, Mita Hospital (Tokyo, Japan). We thank pathologists who have been engaged in reviewing cases and clinicopathological discussion for this study.

## Author Contributions

**Conceptualization:** Masayuki Tsuneki.

**Data curation:** Masayuki Tsuneki, Fahdi Kanavati.

**Formal analysis:** Masayuki Tsuneki, Fahdi Kanavati.

**Investigation:** Masayuki Tsuneki, Fahdi Kanavati.

**Methodology:** Masayuki Tsuneki, Fahdi Kanavati.

**Project administration:** Masayuki Tsuneki.

**Resources:** Masayuki Tsuneki.

**Software:** Masayuki Tsuneki, Fahdi Kanavati.

**Supervision:** Masayuki Tsuneki.

**Validation:** Masayuki Tsuneki, Fahdi Kanavati.

**Visualization:** Masayuki Tsuneki, Fahdi Kanavati.

**Writing – original draft:** Masayuki Tsuneki, Fahdi Kanavati.

**Writing – review & editing:** Masayuki Tsuneki.

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
