## [Decision Letter · Decision Letter 0]

6 Jul 2022

PONE-D-22-13432Weakly supervised learning for multi-organ adenocarcinoma classification in whole slide imagesPLOS ONE

Dear Dr. Tsuneki,

Thank you for submitting your manuscript to PLOS ONE. After careful consideration, we feel that it has merit but does not fully meet PLOS ONE’s publication criteria as it currently stands. Therefore, we invite you to submit a revised version of the manuscript that addresses the points raised during the review process.

ACADEMIC EDITOR

I wish to reinforce the following comments picked up by the peer reviewers: 1) code availability; 2) issues with figures. When evaluating computational pathology submissions, each reviewer is routinely asked to run the provided code and assess its performance independently, which was not possible in this case.

We look forward to receiving your revised manuscript.

Kind regards,

Andrey Bychkov

Academic Editor

PLOS ONE

Journal Requirements:

2. Please provide additional details regarding participant consent. In the Methods section, please ensure that you have specified (1) whether consent was informed and (2) what type you obtained (for instance, written or verbal). If your study included minors, state whether you obtained consent from parents or guardians. If the need for consent was waived by the ethics committee, please include this information

"Fahdi Kanavati and Masayuki Tsuneki are employees of Medmain Inc. (Fukuoka, Japan)." 

We note that one or more of the authors are employed by a commercial company: Medmain Inc.

5.Your ethics statement should only appear in the Methods section of your manuscript. If your ethics statement is written in any section besides the Methods, please move it to the Methods section and delete it from any other section. Please ensure that your ethics statement is included in your manuscript, as the ethics statement entered into the online submission form will not be published alongside your manuscript. 

Reviewers' comments:

Reviewer's Responses to Questions

**Comments to the Author**

1. Is the manuscript technically sound, and do the data support the conclusions?

Reviewer #1: Partly

Reviewer #2: Partly

Reviewer #3: Yes

2. Has the statistical analysis been performed appropriately and rigorously? 

Reviewer #1: No

Reviewer #2: No

Reviewer #3: I Don't Know

3. Have the authors made all data underlying the findings in their manuscript fully available?

Reviewer #1: No

Reviewer #2: No

Reviewer #3: No

4. Is the manuscript presented in an intelligible fashion and written in standard English?

Reviewer #1: Yes

Reviewer #2: Yes

Reviewer #3: Yes

5. Review Comments to the Author

Reviewer #1: The authors use weakly supervised approach for training an algorithm for adenocarcinoma detection.

The study is of potential interest.

There are, however, some important issues:

- Page 2: “(e.g., Hospital-A, B, C, D, and E)”

I would remove word anonymization as this data cannot be anonymized in such way. You can always identify your datasets according to number of slides, staining, cutting peculiarities, etc. What is the meaning of anonymization of hospitals? You probably have to anonymize cases, but not hospitals.

- Page 3: What kind of slides did the authors use from TCGA cohort?

As in Table 5 they state to have 246 slides for lung cancer cohort containing only benign tissue, it is probably the slides of fresh frozen sections. This should be clarified. If authors used fresh frozen sections they should provide the clarification for this as they train on FFPE sections and frozen section is a very different type of material.

- Table 4: Histopathological diagnosis for lymph nodes without metastasis is probably not “Non-neoplastic lesion”, just free of tumor.

- Tumors are extremely heterogenous. How did the pathologists select the cases, how they considered the representativeness. E.g., lung is an organ with a very complex morphology of benign tissue peritumoral. Sometimes you cannot just tell where the tumors starts and where ends. Fibrosis. Inflammation. Not including enough training data can reduce generalizability of the algorithms.

- Authors used EfficientNetB1 as the architecture of neural network. It is a relatively small network and not a “state-of-the-art” network for digital pathology tasks as it is known empirically that larger networks (with 20-50 mln of parameters) might perform better. Importantly, authors refer to their other paper (ref. 24) from where they inherit the technical methodology of the current study. And in ref. 24 they used EfficientNetB3. Given this fact, the authors should clarify their selection and support it by experimental data.

- Authors train only batch normalization layers and the classification layer leaving the whole model untrained (authors should state if they used ImageNet weights or not). This seems strange as in most situations for computational pathology you have to retrain the whole model. Although, ImageNet pretrained models can work as feature extractors on pathology tasks, normally one get up to 5% more accuracy when the whole model is retrained (in transfer learning setting).

- Page 4: “We used k = 8, N = 256, and a batch size of 32”. The selection of exactly these hyperparameters should be clarified and supported by experimental evidence.

- Page 4: Authors start with randomly selecting some patches from training slides (they however do not state, how many patches they use – this should be stated). And train only 1 epoch before switching to hard-mining. By such approach the first selection of random patches can have enormous effects on the result. Authors need to perform some form of bootstrapping or at least present the results from several independent trainings to be able to compare the results among these.

- Page 4: “We used early stopping by tracking the performance of the model on a

validation set; this allows stopping the training when no improvement was observed for more than 10 epochs. The model with the lowest validation loss was chosen as the final model.”

This seems to be unclear for me – validation set is practically several reserved biopsies for each tumor entity. The tumor-bearing biopsies contain regions with tumor and with benign tissue. If the authors did not annotate them, the only way to validate their trained models is on the case/slide level. But the authors state they used validation loss as a trigger for stopping training. How exactly did they calculate loss if they have access only to the validation at case/slide level?

This should be clarified in all details.

- Code availability:

“To train the classification model in this study we adapted the publicly available 140

TensorFlow training script available at https://github.com/tensorflow/models/ 141

tree/master/official/vision/image_classification.”

By this the authors mean that they will not release their code. If it is so, this should be stated clearly and nor referring the readers to the tensorflow repo! At that, this link is not working. However, my personal opinion is that this is inacceptable for research publication in the field of computational pathology because the others have absolutely no chance to reproduce the results. The methodology of the study is very simple so at least at this point there is, in my opinion, no substantial interference with “commercial background” as authors are affiliated to a company. Anyway, if authors do not provide code, there should be robust clarification for this.

- In Table 6 the authors show the test of the existing (ref. 15) model for stomach adenocarcinoma detection on the test datasets of the current study. On the test dataset from the same domain (stomach) AUC is 0.937. It is totally unclear how authors calculated AUC, whether if they used probability or tumor areas thresholds etc, as they make tests on the case/slide level (non-annotated slides). The same is true for log loss.

Authors show AUC = 0.991 for TCGA breast cancer cohort – however this seems to be totally non-believable as this is an absolutely different pathology domain.

- The same issue (exact method for calculation of AUC and log loss, and other metrics is unclear):

“For each test set (stomach endoscopic biopsy, colon endoscopic biopsy, lung TBLB, breast needle biopsy, radical lymph node dissection, lung TCGA, and breast TCGA), we computed the ROC-AUC, log loss, accuracy, sensitivity, and specificity and summarized the results in Table 7 and 8 and Fig. 2B-D.”

- Table 7, Table 8

Again, it is not believable, that you can get AUC 0.999 on such a hard problem like breast cancer training in weakly supervised mode on several hundred biopsies. Probably such results stem from the fact that the authors use only 4 images without tumor for TCGA breast cancer cohort and test at case level. It is a major flaw in design and authors should completely remove TCGA breast dataset from their work.

- Figures: Authors should clarify what different colors mean (probabilities?)

All figures are barely interpretable (the tissue behind color maps cannot be seen)

- Authors used datasets from different institutes. At that they do not adopt any normalization strategies. In this case, different number of cases from different institutes can introduce bias and substantially reduce the generalizability. Authors should ideally provide independent tests using stain normalization strategies. Normally, the results should improve.

- The main achievements of the study (for best models) are presented in Table 8.

Here very low sensitivity and specificity for all tumor entities (TCGA breast should be removed) – actually inacceptable for clinical practice. This reported accuracy corresponds to the inference maps presented in Figures.

This stems probably at least partially from a very low number of cases (selected for study – histologically very complex and heterogeneous tumor entities). It is known from other publication that for weakly supervised approaches 4000 slides are not enough, the number should be definitively > 10 k, ideally 15k-20k slides. Also methodology is sometimes suboptimal: using very small CNN, not training model (!), only classification layers and batch normalization layers. The authors are encouraged to provide the results for training with stain normalization or style transfer, training of full model as well as training in bootstrapping mode (iterations of random selection of training and test cases).

Reviewer #2: The authors present a unique model that is able to classify adenocarcinomas originated from multiple organs in a single model.

Major

1. Breast dataset from TCGA is not appropriate as a test set for cancer detection task, because it is extremely imbalanced (more than 99% are invasive ductal carcinoma).

2. In the heat map in Figs. 3-7, some tiles have frames while others do not, even in areas where the color does not represent the lowest level. In addition, unnatural white to purple grids are seen in the areas without frames. These raise the suspicion of arbitrary image editing. Please provide original images for the evaluation by editor and reviewers and relevant explanations.

3. In validation and test step, how are the inference results at the patch level aggregated to the WSI level diagnosis?

4. Also, please explain how the thresholds for the inference results were set when calculating sensitivity and specificity.

Minor

1. Why TCGA colon cancer and TCGA gastric cancer datasets were not tested?

2. What is the “non-neoplastic lesion” in the TCGA dataset? The TCGA dataset is, by definition, dealing with neoplastic lesions. How the datasets chosen for the test set in this study were selected from the TCGA dataset? To clarify the dataset, the "manifest file" used to download the data from the GDC data portal should be made available.

3. Please explain what cases of metastatic colon cancer of the lungs are in Table 4. Such cases should rarely occur in usual clinical practice.

4. There should be cases where there are multiple WSIs in a single case. How these cases were handled?

5. On which dataset is the deep learning model pretrained?

6. In “code availability” section, please provide author-generated code.

Reviewer #3: The manuscript presents research with important impact and objective, e.g. to provide an auxiliary tool to diagnostic pathology and improve global adenocarcinoma primary screening. I found the approach to be original in aiming to identify adenocarcinoma with considerable accuracy in several primary sites and metastatic lymph nodes, hence matching the diagnostic practice and protocol in multidisciplinary centers with high throughput. Using weakly supervised learning is a strong point in this research foe a list of reasons: Less time in achieving the results without using pathologists annotation efforts, it still remains comprehensive and explainable for the target user , and most importantly achieving very good performance. Below are a list of minor improvements I would suggest :

- The writing used is straight to the point and gets the message across very well, however I would suggest to replace one sentence in the Introduction : Adenocarcinoma is the major cancer arise in these 7

organs, with -Adenocarcinoma is the most common type of cancer affecting these 7 organs.

-Throughout the manuscript is consistently used " histopathological cancer classification" which I understand is linked to the technical realm of computational pathology, but from a practicing pathologist perspective cancer classification goes beyond identification of adenocarcinoma and can be miss interpreted from the reader. I would suggest to replace with " tissue classification and adenocarcinoma detection".

- I appreciated the Method and materials section as it is detailed, well explained and figures show the right data at a glance, however here I was not clear about the validation set size. I found it to be small, but I am not an image analysis expert, and it would be great to add a sentence about the size of validation set criteria if possible.

- It was not entirely clear why the adenomas were pointed out in the class of non-adenocarcinoma for GI ( stomach and colon). I would suggest to explain in a sentence or two the rational for that. e.g. high risk feature , or most common differential etc.

- I would suggest to be more broad when listing the limitations of the study, in addition to the false positives and false negatives. Considering the aim of being impactful in primary screening, the use of only one type of scanner is another limitation to be mentioned for example.

6. PLOS authors have the option to publish the peer review history of their article (what does this mean?). If published, this will include your full peer review and any attached files.

Reviewer #1: No

Reviewer #2: No

Reviewer #3: No

---

## [Author Response · Author response to Decision Letter 0]

18 Jul 2022

Reviewer #1: The authors use weakly supervised approach for training an algorithm for adenocarcinoma detection.

The study is of potential interest.

There are, however, some important issues:

- Page 2: “(e.g., Hospital-A, B, C, D, and E)”

I would remove word anonymization as this data cannot be anonymized in such way. You can always identify your datasets according to number of slides, staining, cutting peculiarities, etc. What is the meaning of anonymization of hospitals? You probably have to anonymize cases, but not hospitals.

Response: We had a set of five hospitals from which we obtained datasets. By anonymising the dataset, we mean that we do not explicitly mention which dataset originated from a given hospital. So as to avoid associating a particular result to an explicitly named hospital.

- Page 3: What kind of slides did the authors use from TCGA cohort?

Response: We’ve expanded the sentence to include that were H&E surgical FFPE specimens

As in Table 5 they state to have 246 slides for lung cancer cohort containing only benign tissue, it is probably the slides of fresh frozen sections. This should be clarified. If authors used fresh frozen sections they should provide the clarification for this as they train on FFPE sections and frozen section is a very different type of material.

Response: We have added a mention that the dataset from TCGA contained both FFPE and frozen sections.

- Table 4: Histopathological diagnosis for lymph nodes without metastasis is probably not “Non-neoplastic lesion”, just free of tumor.

Response: We have replaced all mention of non-neoplastic lesion with non-neoplastic

- Tumors are extremely heterogenous. How did the pathologists select the cases, how they considered the representativeness. E.g., lung is an organ with a very complex morphology of benign tissue peritumoral. Sometimes you cannot just tell where the tumors starts and where ends. Fibrosis. Inflammation. Not including enough training data can reduce generalizability of the algorithms.

Response: The criteria for selection was simply whether the WSI contained adenocarcinoma. Otherwise, it was assumed to be non-neoplastic.

- Authors used EfficientNetB1 as the architecture of neural network. It is a relatively small network and not a “state-of-the-art” network for digital pathology tasks as it is known empirically that larger networks (with 20-50 mln of parameters) might perform better. Importantly, authors refer to their other paper (ref. 24) from where they inherit the technical methodology of the current study. And in ref. 24 they used EfficientNetB3. Given this fact, the authors should clarify their selection and support it by experimental data.

Response: We saw no further improvements from using a larger model, which is why we used the B1 model. In previous study, we have also used B1 [1]

[1] Masayuki Tsuneki and Fahdi Kanavati. "Deep learning models for poorly differentiated colorectal adenocarcinoma classification in whole slide images using transfer learning." Diagnostics 11.11 (2021): 2074.

- Authors train only batch normalization layers and the classification layer leaving the whole model untrained (authors should state if they used ImageNet weights or not). This seems strange as in most situations for computational pathology you have to retrain the whole model. Although, ImageNet pretrained models can work as feature extractors on pathology tasks, normally one get up to 5% more accuracy when the whole model is retrained (in transfer learning setting).

Response: We have added a mention that it is ImageNet pretrained models. Based on extensive results provided in the partial fine-tuning paper [1], it can be sufficient to only fine-tune the affine weights of the batch normalization layers and the final classification layers. That paper includes comparisons of various fine-tuning/transfer learning approaches on histopathology images.

[1] https://proceedings.mlr.press/v143/kanavati21a.html

- Page 4: “We used k = 8, N = 256, and a batch size of 32”. The selection of exactly these hyperparameters should be clarified and supported by experimental evidence.

Response: These parameters had no impact on the final result. They were selected to simply make use of the maximum possible available GPU memory. 

- Page 4: Authors start with randomly selecting some patches from training slides (they however do not state, how many patches they use – this should be stated). And train only 1 epoch before switching to hard-mining. By such approach the first selection of random patches can have enormous effects on the result. Authors need to perform some form of bootstrapping or at least present the results from several independent trainings to be able to compare the results among these.

Response: It is the same parameter k that determines the number of selected patches. We are not proposing a new method here and simply using a method that was validated to work in previous studies. In addition, with deep learning with large training and test datasets, it is the general consensus that it is enough to do a single initial split of the data and perform a single run (See for example Machine Learning Yearning by Andrew Ng), especially when training a single model can take days.

- Page 4: “We used early stopping by tracking the performance of the model on a validation set; this allows stopping the training when no improvement was observed for more than 10 epochs. The model with the lowest validation loss was chosen as the final model.”

This seems to be unclear for me – validation set is practically several reserved biopsies for each tumor entity. The tumor-bearing biopsies contain regions with tumor and with benign tissue. If the authors did not annotate them, the only way to validate their trained models is on the case/slide level. But the authors state they used validation loss as a trigger for stopping training. How exactly did they calculate loss if they have access only to the validation at case/slide level?

Response: This was done on the slide level. The loss is calculated as in any classification problem using binary cross entropy. For the cases in the validation set, we simply applied the model on the entire slide, then took the maximum probability. Now we have a single probability value for the entire slide to compare with the ground truth label. This allows us to compute a loss with the binary cross entropy. We have added a sentence to clarify this.

This should be clarified in all details.

- Code availability:

“To train the classification model in this study we adapted the publicly available 140

TensorFlow training script available at https://github.com/tensorflow/models/ 141

tree/master/official/vision/image_classification.”

By this the authors mean that they will not release their code. If it is so, this should be stated clearly and nor referring the readers to the tensorflow repo! At that, this link is not working. However, my personal opinion is that this is inacceptable for research publication in the field of computational pathology because the others have absolutely no chance to reproduce the results. The methodology of the study is very simple so at least at this point there is, in my opinion, no substantial interference with “commercial background” as authors are affiliated to a company. Anyway, if authors do not provide code, there should be robust clarification for this.

Response: It seems that there have been changes to repo paths, the correct link is https://github.com/tensorflow/models/tree/master/official/vision

At this point, there exists various implementations that validate the weakly supervised training with multiple instance learning. For instance there is a publicly available implementation from a seminal Nature paper here https://github.com/MSKCC-Computational-Pathology/MIL-nature-medicine-2019

We are not proposing a new methodology in this paper. It is simply a clinical application paper, and code is not central to the claim. We do mention that we adapted the code from the tensorflow vision implementation. To avoid any confusion, we have removed this statement.

As far as we are aware, based on the policy of PLOS one, it is not a requirement if the code is not central to the manuscript. https://journals.plos.org/plosone/s/materials-software-and-code-sharing#loc-sharing-code

“In cases where code is central to the manuscript, we may require the code to be made available as a condition of publication. Authors are responsible for ensuring that the code is reusable and well documented”

- In Table 6 the authors show the test of the existing (ref. 15) model for stomach adenocarcinoma detection on the test datasets of the current study. On the test dataset from the same domain (stomach) AUC is 0.937. It is totally unclear how authors calculated AUC, whether if they used probability or tumor areas thresholds etc, as they make tests on the case/slide level (non-annotated slides). The same is true for log loss.

Response: We hope the additional paragraph we added to clarify how the slide-level probabilities are computed also answer this question.

Authors show AUC = 0.991 for TCGA breast cancer cohort – however this seems to be totally non-believable as this is an absolutely different pathology domain.

- The same issue (exact method for calculation of AUC and log loss, and other metrics is unclear):

“For each test set (stomach endoscopic biopsy, colon endoscopic biopsy, lung TBLB, breast needle biopsy, radical lymph node dissection, lung TCGA, and breast TCGA), we computed the ROC-AUC, log loss, accuracy, sensitivity, and specificity and summarized the results in Table 7 and 8 and Fig. 2B-D.”

- Table 7, Table 8

Again, it is not believable, that you can get AUC 0.999 on such a hard problem like breast cancer training in weakly supervised mode on several hundred biopsies. Probably such results stem from the fact that the authors use only 4 images without tumor for TCGA breast cancer cohort and test at case level. It is a major flaw in design and authors should completely remove TCGA breast dataset from their work.

Response: They do in fact stem from only using 4 images without tumor for the TCGA breast cancer cohort. We have removed this dataset from the results.

- Figures: Authors should clarify what different colors mean (probabilities?)

Response: We have this statement mentioned at the end of each figure caption “the heatmap uses the jet color map where blue indicates low probability and red indicates high probability”

All figures are barely interpretable (the tissue behind color maps cannot be seen)

Response: The exact same image on the right contains the tissue without the color map overlay which allows the reader to see the tissue.

- Authors used datasets from different institutes. At that they do not adopt any normalization strategies. In this case, different number of cases from different institutes can introduce bias and substantially reduce the generalizability. Authors should ideally provide independent tests using stain normalization strategies. Normally, the results should improve.

Response: We have included the clarification paragraph that we had performed data augmentation on the brightness, contrast, hue and saturation, which would help account for differences in stain. This had previously been mentioned in the earlier paper that described the method in more detail.

- The main achievements of the study (for best models) are presented in Table 8.

Here very low sensitivity and specificity for all tumor entities (TCGA breast should be removed) – actually inacceptable for clinical practice. This reported accuracy corresponds to the inference maps presented in Figures.

Response: We have removed the breast TCGA dataset.

This stems probably at least partially from a very low number of cases (selected for study – histologically very complex and heterogeneous tumor entities). It is known from other publication that for weakly supervised approaches 4000 slides are not enough, the number should be definitively > 10 k, ideally 15k-20k slides. Also methodology is sometimes suboptimal: using very small CNN, not training model (!), only classification layers and batch normalization layers. The authors are encouraged to provide the results for training with stain normalization or style transfer, training of full model as well as training in bootstrapping mode (iterations of random selection of training and test cases).

Response: We again refer the review to a previous publication on the partial transfer learning method https://proceedings.mlr.press/v143/kanavati21a.html for which code is also available (https://github.com/fk128/batchnorm-transferlearning) to reproduce the results of only fine-tuning the affine parameters of batch normalization layers. We have also added a clarification that we did perform data augmentation on the color to make the model less sensitive to stain.

Reviewer #2: The authors present a unique model that is able to classify adenocarcinomas originated from multiple organs in a single model.

Major

1. Breast dataset from TCGA is not appropriate as a test set for cancer detection task, because it is extremely imbalanced (more than 99% are invasive ductal carcinoma).

Response: We have removed the breast TCGA dataset.

2. In the heat map in Figs. 3-7, some tiles have frames while others do not, even in areas where the color does not represent the lowest level. In addition, unnatural white to purple grids are seen in the areas without frames. These raise the suspicion of arbitrary image editing. Please provide original images for the evaluation by editor and reviewers and relevant explanations.

Response: These images were directly extracted from a web-based viewer based on openseadragon (https://openseadragon.github.io/). The viewer was adapted to display tile overlays of the prediction heatmaps. The predictions were only generated for tissue regions while the rest was automatically assumed to be background. To make tiles easier to visualise at different zoom levels, they had borders added to delineate the tiles to make it easier to view the grid structure of tiles that had prediction. The border lines are proportional to the zoom level. At low magnifications, the border lines are thicker, making them more visible. The more one zooms, the thinner they become, and at maximum magnification, there are no border lines as one can easily see the grid structure of the tiles. This is more of a visual aid. Here is a video showcasing the viewer that shows these borders https://drive.google.com/file/d/167uW8JDOJyfzyRcMP8nxiXx8WuSBmQab/view?usp=sharing

3. In validation and test step, how are the inference results at the patch level aggregated to the WSI level diagnosis?

Response: We have added a paragraph to clarify this.

4. Also, please explain how the thresholds for the inference results were set when calculating sensitivity and specificity.

Response: We used the standard threshold of 0.5.

Minor

1. Why TCGA colon cancer and TCGA gastric cancer datasets were not tested?

Response: We only had the breast and lung TCGA datasets as we had downloaded months prior and reviewed by pathologists. At the time we carried out this study, The TCGA website was not allowing the download of any further images which is why we were unable to obtain other datasets. https://forum.image.sc/t/tcga-slides-not-available-anymore/52532/3

2. What is the “non-neoplastic lesion” in the TCGA dataset? The TCGA dataset is, by definition, dealing with neoplastic lesions. How the datasets chosen for the test set in this study were selected from the TCGA dataset? To clarify the dataset, the "manifest file" used to download the data from the GDC data portal should be made available.

Response: We referred to non-neoplastic lesions as anything that was not neoplastic as reviewed by pathologists. This included things like inflammatory tissue. To further make this clearer we have removed “lesions” so that non-neoplastic simply refers to anything that’s not neoplastic, e.g. non-neoplastic would refer to inflammation and normal.

3. Please explain what cases of metastatic colon cancer of the lungs are in Table 4. Such cases should rarely occur in usual clinical practice.

Response: Here are two representative examples of metastatic colon adenocarcinoma of the lung that we used:

According to the histopathological report, with extensive necrosis, highly atypical columnar epithelial cells proliferate in a fused tubular structure. The patient suffered colon adenocarcinoma and stomach adenocarcinoma as well. Immunohistochemically, the metastatic lesion in lung was positive for CK20 but negative for CK7. In the same patient, stomach adenocarcinoma was positive for CK7 but negative for CK20; and colon adenocarcinoma was positive for CK20 but negative for CK7. Therefore, final diagnosis for the metastatic lesion was “metastatic colon adenocarcinoma of lung”.

According to the histopathological report, highly atypical columnar epithelial cells take on a fused tubular structure and proliferate nodularly with extensive necrosis. Histologically, this is a finding of colorectal cancer metastasis. The same patient suffered colon adenocarcinoma in his/her clinical history.

4. There should be cases where there are multiple WSIs in a single case. How these cases were handled?

Response: We only used a single WSI from a given case.

5. On which dataset is the deep learning model pretrained?

Response: Tables 1 and 2 show which datasets the model was trained on.

6. In “code availability” section, please provide author-generated code.

Response: At this point, there exists various implementations that validate the weakly supervised training with multiple instance learning. For instance there is a publicly available implementation from a seminal Nature paper here https://github.com/MSKCC-Computational-Pathology/MIL-nature-medicine-2019

We are not proposing a new methodology in this paper. It is simply a clinical application paper, and code is not central to the claim. We do mention that we adapted the code from the tensorflow vision implementation. To avoid any confusion, we have removed this statement.

As far as we are aware, based on the policy of PLOS one, it is not a requirement if the code is not central to the manuscript. https://journals.plos.org/plosone/s/materials-software-and-code-sharing#loc-sharing-code

“In cases where code is central to the manuscript, we may require the code to be made available as a condition of publication. Authors are responsible for ensuring that the code is reusable and well documented”

Reviewer #3: The manuscript presents research with important impact and objective, e.g. to provide an auxiliary tool to diagnostic pathology and improve global adenocarcinoma primary screening. I found the approach to be original in aiming to identify adenocarcinoma with considerable accuracy in several primary sites and metastatic lymph nodes, hence matching the diagnostic practice and protocol in multidisciplinary centers with high throughput. Using weakly supervised learning is a strong point in this research foe a list of reasons: Less time in achieving the results without using pathologists annotation efforts, it still remains comprehensive and explainable for the target user , and most importantly achieving very good performance. Below are a list of minor improvements I would suggest :

- The writing used is straight to the point and gets the message across very well, however I would suggest to replace one sentence in the Introduction : Adenocarcinoma is the major cancer arise in these 7

organs, with -Adenocarcinoma is the most common type of cancer affecting these 7 organs.

Response: Done.

-Throughout the manuscript is consistently used " histopathological cancer classification" which I understand is linked to the technical realm of computational pathology, but from a practicing pathologist perspective cancer classification goes beyond identification of adenocarcinoma and can be miss interpreted from the reader. I would suggest to replace with " tissue classification and adenocarcinoma detection".

Response: Done.

- I appreciated the Method and materials section as it is detailed, well explained and figures show the right data at a glance, however here I was not clear about the validation set size. I found it to be small, but I am not an image analysis expert, and it would be great to add a sentence about the size of validation set criteria if possible.

Response: Tables 1 and 2 show the size of the validation sets in parentheses. The test set represents the hold out sets and they’re detailed in Tables 3,4, and 5.

- It was not entirely clear why the adenomas were pointed out in the class of non-adenocarcinoma for GI ( stomach and colon). I would suggest to explain in a sentence or two the rational for that. e.g. high risk feature , or most common differential etc.

Response: We have included it because it’s the most common differential diagnosis as well as to ensure that the model predicts correctly the difference between adenoma and adenocarcinoma due to potential similarity in some features. We have added a sentence to highlight this.

- I would suggest to be more broad when listing the limitations of the study, in addition to the false positives and false negatives. Considering the aim of being impactful in primary screening, the use of only one type of scanner is another limitation to be mentioned for example.

Response: We have added a sentence about the limitations due to the scanner and false positives/negatives.

---

## [Decision Letter · Decision Letter 1]

16 Aug 2022

PONE-D-22-13432R1Weakly supervised learning for multi-organ adenocarcinoma classification in whole slide imagesPLOS ONE

Dear Dr. Tsuneki,

Thank you for submitting your manuscript to PLOS ONE. After careful consideration, we feel that it has merit but does not fully meet PLOS ONE’s publication criteria as it currently stands. Therefore, we invite you to submit a revised version of the manuscript that addresses the points raised during the review process.

We look forward to receiving your revised manuscript.

Kind regards,

Andrey Bychkov

Academic Editor

PLOS ONE

Reviewers' comments:

Reviewer's Responses to Questions

**Comments to the Author**

1. If the authors have adequately addressed your comments raised in a previous round of review and you feel that this manuscript is now acceptable for publication, you may indicate that here to bypass the “Comments to the Author” section, enter your conflict of interest statement in the “Confidential to Editor” section, and submit your "Accept" recommendation.

Reviewer #1: All comments have been addressed

Reviewer #2: (No Response)

2. Is the manuscript technically sound, and do the data support the conclusions?

Reviewer #1: Yes

Reviewer #2: Partly

3. Has the statistical analysis been performed appropriately and rigorously? 

Reviewer #1: Yes

Reviewer #2: Yes

4. Have the authors made all data underlying the findings in their manuscript fully available?

Reviewer #1: No

Reviewer #2: No

5. Is the manuscript presented in an intelligible fashion and written in standard English?

Reviewer #1: Yes

Reviewer #2: Yes

6. Review Comments to the Author

Reviewer #1: Authors addressed most of the points and discussed those they are not able to address.

The manuscript can be accepted in the actual form.

Reviewer #2: Authors removed TCGA breast data this round, however, the data remain in manuscript. Authors should fix them all before submission.

As the reviewer comment before, colon cancer rarely metastasizes to lung lymph node. (Of cause, metastasis to lung is very common in contrast.) These two examples the authors had shown also seem the cases of metastasis to lung (rather than lymph node metastasis). Please discuss again with your consultant pathologists.

It is still unclear what is the “non-neoplastic” slides from TCGA. Authors need to open the lists of cases for both adenocarcinoma and non-neoplastic subgroup, ideally with the inference result for each case. Otherwise, there is no sense in using publicly available data. (Authors did not response the original comment to open manifest file of GDC data portal the authors used for this analysis.)

The result/figure authors claim are all based on an algorithm that authors have developed yourselves. (If there is nothing new in the methodology, then just use the existing code that is publicly available.) There is no doubt that codes are very important in this regard.

PLOS expect all researchers to share author-generated code for reproducibility and reuse. The reviewer hopes the authors to share your code.

7. PLOS authors have the option to publish the peer review history of their article (what does this mean?). If published, this will include your full peer review and any attached files.

Reviewer #1: No

Reviewer #2: No

---

## [Author Response · Author response to Decision Letter 1]

23 Aug 2022

6. Review Comments to the Author

Reviewer #1: Authors addressed most of the points and discussed those they are not able to address.

The manuscript can be accepted in the actual form.

Response: Thank you.

Reviewer #2: Authors removed TCGA breast data this round, however, the data remain in manuscript. Authors should fix them all before submission.

Response: Thank you so much. We found breast TCGA data, so we’ve removed it. The remaining mentions of breast refer to the needle biopsy set that’s not from TCGA.

As the reviewer comment before, colon cancer rarely metastasizes to lung lymph node. (Of cause, metastasis to lung is very common in contrast.) These two examples the authors had shown also seem the cases of metastasis to lung (rather than lymph node metastasis). Please discuss again with your consultant pathologists.

Response: We’ve discussed it again with the pathologists. Though it might be rare, those two examples do correspond to colon cancer metastasis to lung lymph node.

It is still unclear what is the “non-neoplastic” slides from TCGA. Authors need to open the lists of cases for both adenocarcinoma and non-neoplastic subgroup, ideally with the inference result for each case. Otherwise, there is no sense in using publicly available data. (Authors did not response the original comment to open manifest file of GDC data portal the authors used for this analysis.)

Response: We had downloaded the data a while back and imported a copy into our cloud-based viewer. Unfortunately we didn’t keep a copy of the open manifest file. In line with that, we have removed the results as well from the lung TCGA dataset. We have amended Figure 2 by excluding the lung TCGA set.

The result/figure authors claim are all based on an algorithm that authors have developed yourselves. (If there is nothing new in the methodology, then just use the existing code that is publicly available.) There is no doubt that codes are very important in this regard.

PLOS expect all researchers to share author-generated code for reproducibility and reuse. The reviewer hopes the authors to share your code.

Response: We do believe that we are still in line with PLOS policy as the code is not central to the manuscript given that we are not proposing a new methodology and there already exists some publicly available code that implements a similar methodology [1]. 

https://journals.plos.org/plosone/s/materials-software-and-code-sharing#loc-sharing-code

“In cases where code is central to the manuscript, we may require the code to be made available as a condition of publication. Authors are responsible for ensuring that the code is reusable and well documented”

[1] https://github.com/MSKCC-Computational-Pathology/MIL-nature-medicine-2019

---

## [Decision Letter · Decision Letter 2]

12 Sep 2022

PONE-D-22-13432R2Weakly supervised learning for multi-organ adenocarcinoma classification in whole slide imagesPLOS ONE

Dear Dr. Tsuneki,

Thank you for submitting your manuscript to PLOS ONE. We invite you to submit a revised version of the manuscript that addresses the minor points raised during the review process.

We look forward to receiving your revised manuscript.

Kind regards,

Andrey Bychkov

Academic Editor

PLOS ONE

Journal Requirements:

Reviewers' comments:

Reviewer's Responses to Questions

**Comments to the Author**

1. If the authors have adequately addressed your comments raised in a previous round of review and you feel that this manuscript is now acceptable for publication, you may indicate that here to bypass the “Comments to the Author” section, enter your conflict of interest statement in the “Confidential to Editor” section, and submit your "Accept" recommendation.

Reviewer #2: All comments have been addressed

2. Is the manuscript technically sound, and do the data support the conclusions?

Reviewer #2: Yes

3. Has the statistical analysis been performed appropriately and rigorously? 

Reviewer #2: Yes

4. Have the authors made all data underlying the findings in their manuscript fully available?

Reviewer #2: No

5. Is the manuscript presented in an intelligible fashion and written in standard English?

Reviewer #2: Yes

6. Review Comments to the Author

Reviewer #2: 

- Authors need to clearly state the major limitation in the Discussion that the models were not validated in independent cohort(s) from different institutions.

- Fix in in the Abstract that the models were evaluated in five (not seven) independent test sets.

7. PLOS authors have the option to publish the peer review history of their article (what does this mean?). If published, this will include your full peer review and any attached files.

Reviewer #2: No

---

## [Author Response · Author response to Decision Letter 2]

14 Sep 2022

Journal Requirements:

Response: It is complete and correct.

Reviewers' comments:

6. Review Comments to the Author

Reviewer #2: 

- Authors need to clearly state the major limitation in the Discussion that the models were not validated in independent cohort(s) from different institutions.

Response: We have added this as the last sentence in the discussion.

- Fix in in the Abstract that the models were evaluated in five (not seven) independent test sets.

Response: Fixed. Thank you so much.

---

## [Editor Report · Decision Letter 3]

15 Sep 2022

Weakly supervised learning for multi-organ adenocarcinoma classification in whole slide images

PONE-D-22-13432R3

Dear Dr. Tsuneki,

We’re pleased to inform you that your manuscript has been judged scientifically suitable for publication and will be formally accepted for publication once it meets all outstanding technical requirements.

Kind regards,

Andrey Bychkov

Academic Editor

PLOS ONE

---

## [Editor Report · Acceptance letter]

21 Sep 2022

PONE-D-22-13432R3 

Weakly supervised learning for multi-organ adenocarcinoma classification in whole slide images 

Dear Dr. Tsuneki:

I'm pleased to inform you that your manuscript has been deemed suitable for publication in PLOS ONE. Congratulations! Your manuscript is now with our production department. 

Kind regards, 

on behalf of

Dr. Andrey Bychkov 

Academic Editor

PLOS ONE